# VIDEO-TO-AUDIO GENERATION WITH HIDDEN ALIGNMENT

## ABSTRACT

Generating semantically and temporally aligned audio content in accordance with video input has become a focal point for researchers, particularly following the remarkable breakthrough in text-to-video generation. In this work, we aim to offer insights into the video-to-audio generation paradigm, focusing on three crucial aspects: vision encoders, auxiliary embeddings, and data augmentation techniques. Beginning with a foundational model built on a simple yet surprisingly effective intuition, we explore various vision encoders and auxiliary embeddings through ablation studies. Employing a comprehensive evaluation pipeline that emphasizes generation quality and video-audio synchronization alignment, we demonstrate that our model exhibits state-of-the-art video-to-audio generation capabilities. Furthermore, we provide critical insights into the impact of different data augmentation methods on enhancing the generation framework's overall capacity. We showcase possibilities to advance the challenge of generating synchronized audio from semantic and temporal perspectives. We hope these insights will serve as a stepping stone toward developing more realistic and accurate audio-visual generation models.

## 1 INTRODUCTION

We humans perceive visual and audio input as two complementary aspects of our surroundings. When we witness an event, we instinctively expect to hear the corresponding sounds as we see it unfold. This integrated perception of sight and sound enhances our understanding and interpretation of the world around us. Modern video generation models, however, typically focus on generating visual content based on a set of textual prompts (Blattmann et al., 2023; Zeng et al., 2023; Bar-Tal et al., 2024). These models often lack the ability to incorporate audio information, resulting in outputs that appear as inferior imitations rather than realistic representations.

Our objective is to generate semantically and temporally-aligned audio content for a given silent video. This ambitious goal is rooted in extensive research conducted in the text-to-audio (TTA) domain, where researchers have proposed several robust baselines for audio generation (Kreuk et al., 2022; Liu et al., 2023a; Ghosal et al., 2023). Furthermore, there have been recent attempts at video-to-audio (VTA) tasks (Sheffer & Adi, 2023; Luo et al., 2024; Wang et al., 2024). While these works have made significant strides, they still fall short of natural and arbitrary audio generation.

A common trait amongst these baseline models is their utilization of non-autoregressive generation frameworks, particularly Latent Diffusion Model (LDM) (Rombach et al., 2022). These models generate audio content by leveraging textual or visual features as generative conditions. However, the distinction between VTA and TTA presents two key challenges: 1) ensuring semantic coherence with the input condition, and 2) ensuring temporal alignment between the generated audio and the video. This is due to the necessity of accurately extracting and interpreting visual features that produce sounds, as well as identifying when these features generate sounds, even if they are continuously present in the scene.

We try to provide insights into the VTA training paradigm by focusing on three key aspects: 1) the vision encoder, 2) the auxiliary embedding, and 3) the data augmentation. Vision encoder is responsible for extracting and interpreting visual features from the input video, capturing complex visual patterns that are integral to generating relevant audio content. It also has the potential to encode temporal information. Auxiliary embedding serves as a valuable source of additional contextual

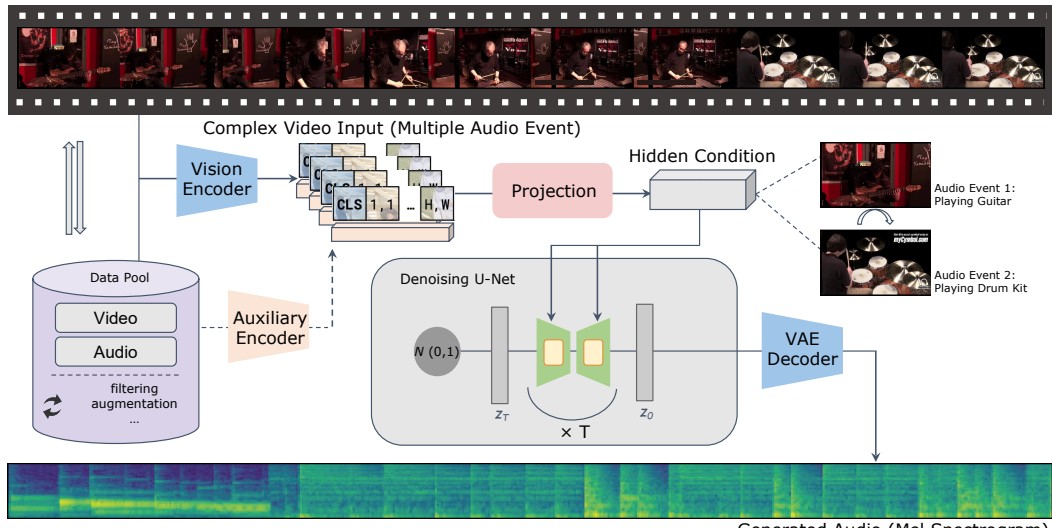

Figure 1: **Overview of the VTA-LDM framework.** Given the silent video, our model generates semantically-related and temporally-aligned audios that accurately correspond to the visual events. The framework is based on a LDM with encoded vision features as the generation condition.

information for the model, e.g. textual description, position embedding, or other metadata associated with the video. Data augmentation can help improve the model's generalization capabilities by introducing variations and perturbations in the training data. In particular, time-stretching of the training data can create fast-switching video scenes, which can push the model to better learn and understand temporal relations between visual and auditory information.

Our study initiates from training a foundational VTA model, denoted as VTA-LDM, drawing upon the successful text-to-audio LDM framework. By employing a Clip-based vision encoder (Radford et al., 2021), we concatenate frame-level video features temporally and map them using a projector as the generation condition. The overall framework of VTA-LDM is shown in fig. 1. The evaluation of these models concentrates on two primary aspects: semantic alignment and temporal alignment. Semantic alignment assesses the generated output's semantic coherence and audio relevance, while temporal alignment gauges the synchronization between the generated audio and corresponding video frames. We then conduct comprehensive ablation experiments to examine the three key factors that can influence the overall performance. Results show that even our basic framework achieves state-of-the-art results in TTA tasks, generating semantically and partly-temporally aligned audio content based on visual input. Furthermore, the integration of additional features into the framework can significantly enhance the generation quality and the synchronization between audio and visual elements from various perspectives. For instance, the inclusion of extra textual prompts can aid in semantic understanding, positional embedding and concatenation augmentation can improve temporal alignment, and pretraining can enhance the overall quality of generation. These findings indicate that the overall framework empowers the model to learn and comprehend the dynamics of the scene and demonstrates the potential for further exploration and refinement.

1. We introduce an effective VTA framework, named VTA-LDM, which attains state-of-the-art performance in VTA tasks.

2. We investigate three fundamental aspects within the VTA paradigm, providing valuable insights into model design and training processes: 1) a linear combination of semantic video features is sufficient for encoding information required for aligned audio generation; 2) additional auxiliary embeddings assist in filtering out chaotic visual information and focusing on crucial elements; 3) pre-training enhances overall generation capability, and filtering high-quality data improves audio-video alignment.

3. Through numerous ablation experiments and evaluations, we supply detailed quantitative results for VTA tasks.

We hope our experiments and results lay a solid foundation for future research in VTA field.

## 2 RELATED WORK

### 2.1 DIFFUSION-BASED GENERATION

The success of audio generation tasks primarily stems from TTA diffusion models. These diffusion models were initially employed in image generation tasks (Ho et al., 2020; Dhariwal & Nichol, 2021; Nichol et al., 2021; Rombach et al., 2022), and then extended to audio and video generation (Liu et al., 2023a;b; Singer et al., 2022; Blattmann et al., 2023; Brooks et al., 2024). Compared with previous works like Audiogen (Kreuk et al., 2022), which are predominantly based on regressive generative models, diffusion-based models excel in generation quality and diversity. For instance, Audioldm (Liu et al., 2023a) learns continuous audio representations in the latent space. Tango (Ghosal et al., 2023) incorporates an instruction-tuned Large Language Model (LLM) FLAN-T5 to leverage its powerful representational capabilities. Other studies (Liu et al., 2023b; Huang et al., 2023; Luo et al., 2024; Zeng et al., 2023; Xu et al., 2024) also utilize diffusion models, in conjunction with other enhancing methods, to improve the generation capacity.

### 2.2 MULTIMODAL AUDIO GENERATION

Numerous studies have concentrated on multi-modal generation tasks related to the audio modality, extending beyond text and conducting significant experiments across various modalities. Previous research (Iashin & Rahtu, 2021), has focused on multi-class visually guided sound synthesis utilizing VQGAN (Esser et al., 2021). Recent work includes IM2WAV (Sheffer & Adi, 2023), which employs a two-stage audio generation pipeline to generate low-level audio representation autoregressively, followed by high-level audio tokens from images. Similarly, Diff-Foley (Luo et al., 2024) uses contrastive audio-visual pretraining to learn temporally aligned features for synthesizing audio from silent videos, drawing on LDM. Seeing&Hearing (Xing et al., 2024) and T2AV (Mo et al., 2024) show open-domain visual-audio generation capability based on similar multi-modality latent aligners. Other studies (Kurmi et al., 2021; Ruan et al., 2023; Yariv et al., 2024) have attempted joint video-audio generation using the same or aligned latent. Despite these advancements, generating synchronized audio that corresponds with a given video remains a significant challenge, particularly at the audio event level.

## 3 BASELINE FRAMEWORK

### 3.1 OVERALL FRAMEWORK

Drawing inspiration from previous TTA works (Liu et al., 2023a; Ghosal et al., 2023) and multi-modal research (Luo et al., 2021; Sheffer & Adi, 2023), we develop a fundamental VTA framework VTA-LDM consisting of several key components: a vision encoder, a conditional LDM, and a mel-spectrogram/audio variational auto-encoder (VAE). Specifically, we utilize vision features extracted from pre-trained vision encoders and feed them to the LDM as the generation condition through a linear projection layer. The LDM operates on the latent audio representation of the mel-spectrogram. The pre-trained audio VAE assists in decoding the denoised latent output to a mel-spectrogram, which is then fed to a vocoder to generate the final audio.

### 3.2 VISION ENCODER

Vision Encoder is responsible for encoding not only the semantic meanings of the video $V$ but also the temporal information required for alignment with the generated audio. We employ pretrained vision encoders $f_V$ such as CLIP4CLIP (Luo et al., 2021) to extract the visual features from the input video. These features capture the essential visual information, including objects, actions, and scene context. We use a projection layer $\phi$ to map these features to the desired dimension of the diffusion condition. The vision encoders are all frozen during the training process, while the projection layer will be trained from scratch.

### 3.3 LATENT DIFFUSION MODEL (LDM)

Given the original input $x_0$, diffusion models follow a Markov chain of diffusion steps to gradually add random noise to data until it follows a Gaussian distribution $N(0, I)$, represented by $x_t$. The model then learns the reverse de-noising process to recover the original input data. For computational efficiency, LDM incorporates a trained perceptual compression model to encode the input $x$ into a low-dimensional latent space $Z$. In text-based generation tasks, the generation condition is often a given text description. In our VTA implementation, the condition $c$ is the projected video feature $\phi(f_V)$. To perform sequential de-noising, we train a network $\epsilon_\theta$ to predict artificial noise, following the objective:

$$\min_\theta \mathbb{E}_{z_0, \epsilon \sim \mathcal{N}(0,I), t \sim \text{Uniform}(1,T)} \|\epsilon - \epsilon_\theta(z_t, t, \phi(f_V))\|_2^2, \tag{1}$$

where the condition $C = \phi(f_V)$ is the projected visual embedding of the input video. We employ a classifier-free generation approach in the reverse process, which has been shown to be an effective method for text-guided generation and editing Ho & Salimans (2022). Given a latent variable and a textual prompt, the generation is performed both conditionally and unconditionally, and then weighted according to a given scale. Formally, let $\varnothing$ be the null text embedding, $w$ denotes the guidance scale, and the generation process can be defined by:

$$\tilde{\epsilon}_\theta = w \cdot \epsilon_\theta(z_t, t, \phi(f_V)) + (1 - w) \cdot \epsilon_\theta(z_t, t, \varnothing). \tag{2}$$

In text-based generation, $\varnothing$ is commonly defined as $\phi(" ")$ to represent the null condition. In TTA tasks, we use a zero embedding to serve as the $\phi(NULL)$, representing the null visual condition.

### 3.4 AUDIO VAE AND VOCODER

The audio source is firstly conveyed to the mel-spectrogram using Short-Time Fourier Transform (STFT), and then to the latent representation $z$ with a pre-trained audio VAE. We use the pre-trained VAE from AudioLDM (Liu et al., 2023a). In the reverse process, the audio is recovered from the latent $z$ using the Hifi-GAN vocoder (Kong et al., 2020).

## 4 EXPERIMENTAL SETUP

### 4.1 DATASET

Our primary experiments are conducted on VGGSound Chen et al. (2020), a dataset comprising over 550 hours of videos with acoustic visual-audio event pairs. We train our models on 200k videos and evaluate them on 3k videos. For certain ablations, we either construct an augmented test set based on the original dataset by randomly cutting and combining segments, or import large unlabelled audio or video corpora for pre-training purposes. More details regarding the dataset preparation, training, and evaluation procedures can be found in section 5.4 and section 5.5.

### 4.2 OBJECTIVE EVALUATION

We utilize commonly used quantitative metrics for the objective evaluation. For evaluating semantic consistency, Fréchet distance (FD), Fréchet audio distance (FAD), Inception Score (IS), Kullback–Leibler (KL) measure the semantic similarity of the generated audios with the ground-truth target in distribution, as well as the generation diversity and quality of the audios. Contrastive language-audio pretraining (CLAP) (Wu* et al., 2023) score calculates how well the generated audio aligns with the textual description of the audio. For evaluating temporal alignment, AV-Align (Yariv et al., 2024) assesses the alignment of generated audio with the input video. CAVP (Luo et al., 2024) evaluates how well the audio aligns with the visual input at both semantic and temporal levels. Besides, Prompting Audio-Language Models (PAM) (Deshmukh et al., 2024) assesses the quality of generated audio.

### 4.3 SUBJECTIVE EVALUATION

We include a subjective evaluation of our models, assessing overall quality, audio quality, video-audio semantic alignment, and video-audio temporal alignment using a scale of 1 to 100. In the

subjective evaluation, we combine the generated audio with the original video pieces, and show the combined video to the human participants without telling them the concrete test cases. Detailed definition of these metrics can be found in appendix B.

# 5 EXPERIMENTS

## 5.1 BASIC CONFIGS

Our LDM structure is configured with a cross-attention dimension of 2048 and features 8 input and output channels. We train our models with 300 warmup steps, a learning rate of 6e-5, and a batch size of 128. All models are trained for a total of 120 epochs. All the models are trained on 8 Nvidia A100 cards. During the inference steps, we set the denoise steps to 300, and the number of samples per audio 1. We utilize classifier-free guidance with a guidance scale of 3.

## 5.2 INITIAL RESULTS

We first compare our foundational model with other existing VTA baselines to show its capability in audio generation, including IM2WAV (Sheffer & Adi, 2023), Diff-Foley (Luo et al., 2024), Foley-Crafter (Zhang et al., 2024), Seeing&Hearing (Xing et al., 2024) and T2AV (Mo et al., 2024). These compared baselines primarily utilize the diffusion framework. Although there are additional VTA baselines available, we consider these to be representative and indicative of SOTA performance. We also list the vision encoders these baselines use for reference. For certain benchmarks, we omit their alignment metrics, as they are tailored to produce shorter audio segments, which does not align with competitive standards. See appendix A for more details about these baselines.

Table 1: **Comparison with other baselines.**

| | VE | FAD ↓ | IS ↑ | FD ↓ | KL ↓ | PAM ↑ | CLAP ↑ | CAVP ↑ | AV-Align ↑ |
|---|---|---|---|---|---|---|---|---|---|
| GT | - | - | - | - | 0.316 | 0.492 | 0.813 | 0.237 | |
| IM2WAV | CLIP | 6.32 | 7.46 | 42.13 | 2.38 | 0.182 | 0.306 | 0.781 | 0.192 |
| Diff-Foley | CAVP | 7.32 | 9.62 | 41.09 | 6.03 | 0.226 | 0.441 | **0.802** | 0.187 |
| FoleyCrafter | CLIP | 4.44 | 9.44 | 27.00 | 4.57 | **0.307** | 0.179 | 0.762 | 0.239 |
| Seeing&Hearing | ImageBind | 7.32 | 5.83 | 32.92 | 2.62 | - | - | - | - |
| T2AV | VA-CLAP | 4.05 | 8.02 | 33.29 | **2.12** | - | - | - | - |
| VTA-LDM (Ours) | CLIP4Clip | **2.05** | **10.10** | **25.50** | 3.81 | 0.245 | **0.452** | 0.800 | **0.243** |

Table 1 presents the results of our foundational model compared to existing VTA baselines with different vision encoders (VE), demonstrating its capability in audio generation[1]. We observe that our model is competitive with all of the baselines across the metrics, showcasing its superiority in generating high-quality, diverse, and temporally-aligned audios. Specifically, our model achieves a lower FAD and FD compared to the IM2WAV and Diff-Foley baseline, indicating better distribution similarity between generated and ground truth audios. Regarding CLAP, CAVP, and AV-Align, our model exhibits superior performance compared to the baselines, emphasizing its ability to generate semantically-related and temporally-aligned audios.

## 5.3 EXPERIMENTS ON VISION ENCODERS

We begin by investigating how different vision encoders may influence the VTA results. Our study includes popular vision encoders such as Clip4Clip (Luo et al., 2021), Imagebind (Girdhar et al., 2023), LanguageBind (Zhu et al., 2023), V-JEPA (Bardes et al., 2024), ViViT (Arnab et al., 2021), and CAVP (Luo et al., 2024). These vision encoders can primarily be categorized into two groups: 1) image encoders trained with multimodal alignment to enhance semantic understanding (*e.g.*, Clip4Clip and Imagebind), and 2) video feature encoders designed to extract semantic and temporal information from videos. Details can be found in appendix C.

We show the results in table 2. The experiments highlight the influence of different vision encoders on the video-to-audio generation task. The choice of vision encoder plays a critical role in capturing

---

[1]For Seeing&Hearing and T2AV, as the codes are not publicly released, we adopt the metrics from their original papers.

Table 2: **Ablation on vision encoders.**

|  | FAD ↓ | IS ↑ | FD ↓ | KL ↓ | PAM ↑ | CLAP ↑ | CAVP ↑ | AV-Align ↑ |
|---|---|---|---|---|---|---|---|---|
| Clip4Clip(VTA-LDM) | **2.05** | **10.10** | **25.50** | **3.81** | 0.245 | **0.452** | **0.800** | **0.225** |
| ImageBind | 8.40 | 6.85 | 42.18 | 8.41 | 0.166 | 0.186 | 0.791 | 0.189 |
| LanguageBind | 2.94 | 9.03 | 28.65 | 4.76 | 0.201 | 0.338 | 0.798 | 0.200 |
| ViViT | 15.99 | 3.88 | 58.14 | 7.33 | **0.288** | 0.223 | 0.766 | 0.189 |
| V-JPEA | 14.28 | 4.27 | 60.04 | 11.89 | 0.142 | 0.080 | 0.786 | 0.186 |
| CAVP | 4.27 | 4.83 | 56.77 | 3.88 | 0.192 | 0.137 | 0.787 | 0.220 |

the semantic and temporal information from the input videos, which directly impacts the quality, diversity, and alignment of the generated audio. From the results, it is evident that Clip4Clip, which is based on the pre-trained model Clip, outperforms other methods in generating high-quality and diverse semantically-related audio content. On the other hand, methods like ViViT and V-JPEA, which utilize different vision encoders, show comparatively lower performance in most metrics. While in these cases, CAVP achieves better temporal alignment as shown by the AV-Align score and the CAVP score.

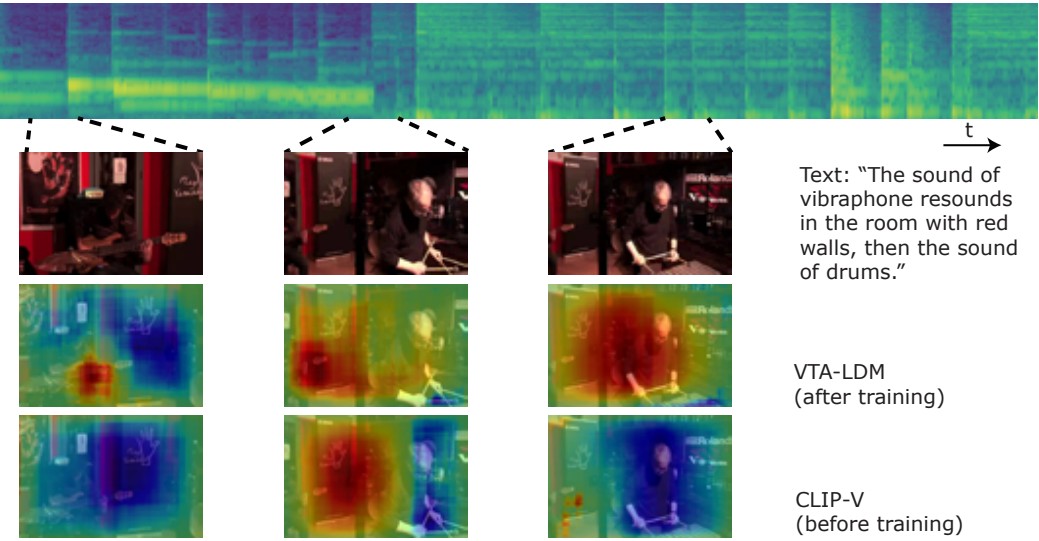

Figure 2: **The saliency map of our model's interest upon the visual input.** We illustrate that VTA-LDM has the ability to learn and concentrate on potential objects capable of producing sound. Furthermore, the model is designed to focus on various sections of the frame across different time intervals, although the attention is calculated based on the final audio latent representation only once.

To further demonstrate how the model's generation part engagements with the input video, we compute the saliency map of the input visual feature based on the generated audio content, aiming to empirically show how VTA-LDM processes the input video frames. As illustrated in fig. 2, our model focuses on different potential video regions that could potentially be the source of the audio. The model can engage in an attention shuffle with the frames as they progress, indicating its ability to capture temporal information between the frames, despite using the pure semantic vision encoder CLIP. For a safe conclusion, we note that a simple temporal combination of semantic vision features of the video frames may be sufficient to capture the features from the video in VTA tasks, eliminating the need for re-training a video feature encoder.

## 5.4 EXPERIMENTS ON AUXILIARY EMBEDDINGS

Auxiliary embeddings serve as additional information beyond visual features, potentially enhancing video understanding and anchoring audio events to improve the generation process. We explore the integration of extra information from various modalities, including textual descriptions that en-

capsulate the overall video content, position embeddings that indicate the sequence of frames, and optical flow that captures pixel-level changes between frames within the video. Details can be found in appendix D.

Table 3: **Ablation on auxiliary embeddings.**

| | FAD ↓ | IS ↑ | FD ↓ | KL ↓ | PAM ↑ | CLAP ↑ | CAVP ↑ | AV-Align ↑ |
|---|---|---|---|---|---|---|---|---|
| Text-only | **1.98** | 10.02 | 21.25 | **2.39** | 0.278 | **0.486** | 0.764 | 0.176 |
| Vision-only(VTA-LDM) | 2.05 | 10.10 | 25.50 | 3.81 | 0.245 | 0.452 | 0.800 | 0.225 |
| Vision+Text | 2.09 | 10.61 | **21.13** | 2.47 | **0.284** | **0.486** | **0.802** | **0.228** |
| Vision+PE | 2.36 | 8.16 | 29.71 | 4.88 | 0.250 | 0.332 | 0.801 | 0.223 |
| Vision+Text+PE | 2.99 | **12.71** | 25.98 | 2.76 | 0.280 | 0.476 | 0.794 | 0.189 |
| Vision+Optical Flow | 3.02 | 9.88 | 28.32 | 3.01 | 0.230 | 0.473 | 0.792 | 0.224 |

Results in table 3 show that auxiliary embeddings, such as extra text embeddings and extra position embeddings, provide additional information that can enhance the model's understanding of the input data and improve the quality of generated audios. Adding extra text embeddings improves the Inception Score (IS) from 10.10 to 10.61 and reduces the FAD, FD, and KL, indicating that the additional semantic information from the text can enhance the quality and diversity of the generated audios, while adding extra position embeddings (PE) appears to have a mixed effect. Combining both text and position embeddings yields the highest IS score of 12.71 but also increases the FAD and reduces the AV-Align score. This suggests that while the combined embeddings can enhance the diversity of the generated audios, they might not necessarily improve the alignment with the ground truth audios.

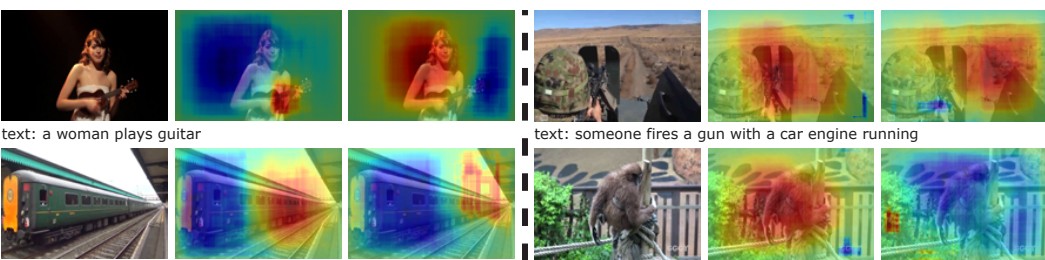

Figure 3: **A Comparison Between Models Without Additional Text Embedding.** The left saliency maps encode the text embedding, while the right ones do not. We demonstrate that extra text embeddings can aid the model in gaining a deeper understanding of the visual content.

We observe that additional embeddings, particularly extra text embeddings, can not only assist the model in comprehending the input video but also concentrate on the actual object producing the audio, thereby improving audio-visual alignment, as illustrated in fig. 3. Visual inputs are more complex and chaotic, making it challenging to discern the true source of the audio. For instance, (on the top left) a woman speaks without moving her mouth, but a non-fine-grained vision encoder cannot recognize this. The extra textual input can help identify the real focus, which is the "guitar."

## 5.5 EXPERIMENTS ON DATA AUGMENTATION

Table 4: **Ablation on data augmentation.**

| | FAD ↓ | IS ↑ | FD ↓ | KL ↓ | PAM ↑ | CLAP ↑ | CAVP ↑ | AV-Align ↑ |
|---|---|---|---|---|---|---|---|---|
| None(VTA-LDM) | **2.05** | 10.10 | 25.50 | 3.81 | 0.245 | 0.452 | 0.800 | 0.225 |
| Data Alignment Filter | 3.85 | 8.19 | 33.43 | **0.95** | **0.314** | **0.484** | **0.802** | **0.281** |
| Concat Augment | 2.47 | 10.82 | 25.09 | 3.59 | 0.236 | 0.393 | 0.798 | 0.248 |
| Pretrain-Video | 2.93 | 9.90 | 32.10 | 1.28 | 0.292 | 0.267 | 0.769 | 0.262 |
| Pretrain-Audio | 2.48 | **12.23** | **23.61** | 1.38 | 0.232 | 0.469 | 0.798 | 0.250 |

The data used in training is also crucial. We explore several approaches from different perspectives. Initially, we filter the training set using video-audio alignment labels to obtain high-quality video-audio pairs. We use a CLAP (Wu* et al., 2023) model to help select audio-video pairs with similar semantics based on extra textual labels (with $score > 0.3$). We also use the AV-Align score to filter unmatched video-audio pairs (with $score > 0.2$). Furthermore, we utilize data augmentation techniques to enhance the diversity and complexity of the training set. We randomly combine video and audio segments to construct a diverse and multi-content set of training data points. Additionally, we leverage extra training sets for pretraining and perform fine-tuning on the foundational model. These approaches are separately studied to illustrate their influence on the final generation results. More details can be found in appendix E.

The influence of different data augmentation methods on the performance of the VTA generation model can be observed in table 4. We compare the base model without augmentation to models utilizing various data augmentation techniques. Data Clean leads to the most significant performance improvement. Removing noisy or irrelevant video-audio samples results in improvements mainly in audio quality and video-audio alignment. The drop in generation quality can be attributed to the decrease in the number of training instances. Concat Augment leads to an improvement in IS, suggesting that this technique can enhance the diversity of the generated audio content, as well as a significant improvement in AV-Align, indicating better video-audio alignment. However, the distance metrics, such as FAD and KL, show a slight decrease with these data augmentation methods. It is worth noting that while data augmentations within the data corpus can improve diversity and alignment, there may be a trade-off in terms of audio content itself. Pretraining with extra data is a natural idea that can help address this issue. The results show that pretraining on extra unlabelled video and audio data can help improve some generation metrics. We believe an appropriate combination of these augmentation methods can enhance the overall performance of the VTA task.

# 6 FURTHER EXPERIMENTS & DISCUSSION

## 6.1 SUBJECTIVE EVALUATION

Table 5: **Subjective evaluation results.**

|  | Audio Quality ↑ | Semantic Alignment ↑ | Temporal Alignment ↑ | Overall Quality ↑ |
|---|---|---|---|---|
| GT | 91.06±8.87 | 91.14±12.54 | 89.66±14.04 | 91.06±10.62 |
| Diff-Foley | 48.45±11.44 | 47.03±28.45 | 57.17±28.20 | 46.26±19.72 |
| IM2WAV | 75.01±16.77 | 58.35±28.94 | 48.33±20.85 | 54.45±21.59 |
| VTA-LDM | 72.52±20.71 | 64.92±24.51 | 43.50±25.52 | 49.04±22.23 |
| VTA-LDM+Text | 83.30±11.45 | 69.45±22.99 | 59.97±26.31 | 66.12±21.57 |
| VTA-LDM+Text+Concat | **86.46**±11.48 | **83.59**±16.69 | **79.32**±17.64 | **82.06**±15.52 |

We selected several ablation models with relatively better performance for the subjective evaluation to compare against the ground truth and other baselines. In each group, we randomly selected 20 video-audio pairs. The results are presented in Table 5.

In general, we observe that the subjective evaluations from our human participants align with the objective metrics, where our framework outperforms the other baselines. However, we also note that the performance still has room for improvement to reach the level of natural and realistic audio. Upon closer examination of the individual metrics, we observe that the VTA-LDM+Text+Concat model yields the best results among the ablation models, and also surpasses other baselines, indicating the effectiveness of combining auxiliary embeddings and data augmentation techniques. The improvements in semantic and temporal alignment, in particular, highlight the importance of incorporating these elements in the video-to-audio training paradigm.

We also note that almost all baselines exhibit a significant variance. Upon further examination of the generated cases, we find that the models are sensitive to certain factors, such as the complexity of the video scene, the presence of multiple audio sources. From any level, there is still a noticeable gap between the performance of existing VTA models and the ground truth.

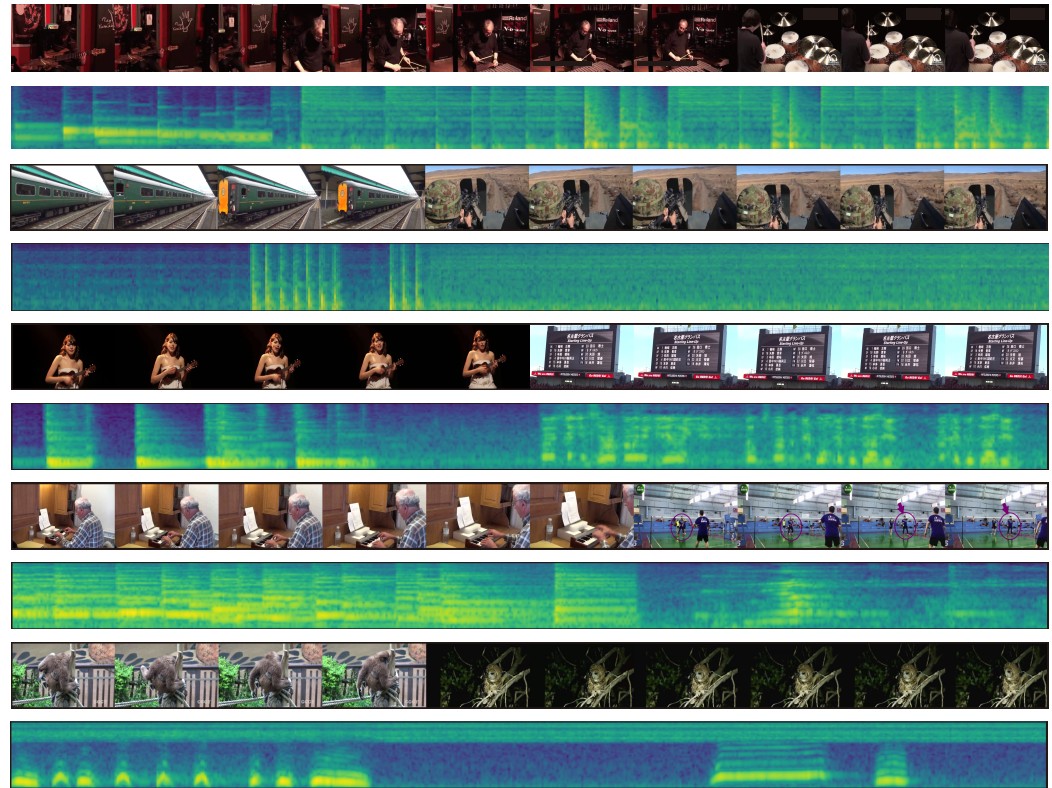

Figure 4: **Demos of the VTA generation.** Given the silent video, our model generates semantically-related and temporally-aligned audios that accurately correspond to the visual events.

## 6.2 OVERALL DISCUSSION

We show a branch of TTA demos in fig. 4. Based on the quantitative results above, we can safely conclude that our vanilla structure has demonstrated its capability to effectively address the semantic alignment of TTA tasks. The vanilla structure's design enables it to capture and translate the semantic information from the text prompts into the generated audio content, resulting in high-quality, semantically-aligned audio outputs.

However, while the vanilla structure exhibits the potential to address the temporal alignment of TTA tasks, it only partially solves this problem. Temporal alignment, which involves aligning the generated audio events with the corresponding visual events in the video, is a more complex issue that requires additional considerations. We introduced additional modifications, such as position encoding and data augmentation, to enhance the overall performance. Position encoding helps the model better understand the temporal order of events, improving the temporal alignment of the generated audio content. Data augmentation, on the other hand, enhances the model's generalizability by exposing it to a wider variety of complex data scenarios. While these modifications have generally led to improvements in both semantic and temporal alignment, they have also occasionally resulted in negative feedback, as shown in table 3 and table 4.

Figure 4 shows that our model generally performs well in temporally generating corresponding audios in several complex scenarios. However, there are instances where the model slightly miscatch the change of the scene. From our point of view, several key problems cannot be solved by the current technique, which may be ignored in the previous study: 1) the presence of an object in a video doesn't necessarily equate to it producing a sound. Some objects can produce different sounds depending on the context or the action being performed, and not all objects in a video make a sound. Models often fail to recognize these 'silent' objects in a video. Nevertheless, our model shows robust generation capabilities in open-domain tasks, generating natural and realistic audios from silent YouTube or even AI-generated videos, as shown in fig. 5.

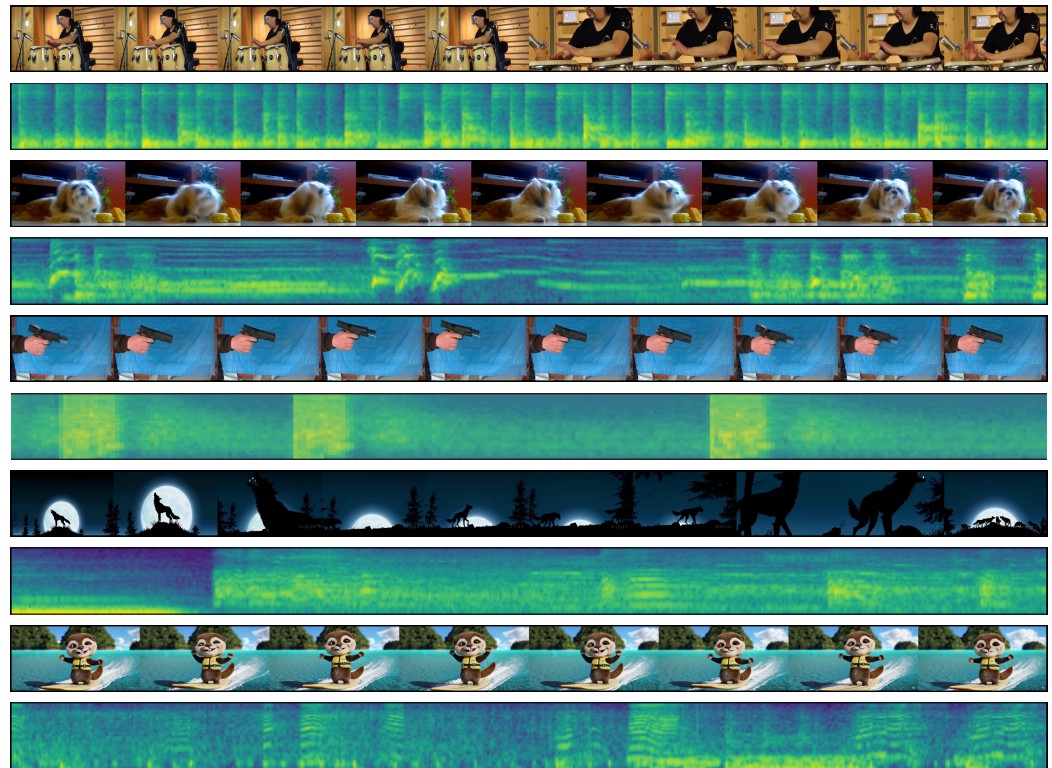

Figure 5: **Demos of the VTA generation on open-domain videos.** Videos are collected from YouTube or generated by OpenAI Sora (Brooks et al., 2024). Although some test data points may exhibit styles distinct from those in the training dataset, our model concentrates on the semantic comprehension of video content and possesses a certain degree of out-of-domain generalization capability.

## 7    CONCLUSION

In this paper, we delve into the VTA task, which aims to generate semantically-related and temporally-aligned audios given silent video pieces. Leveraging the diffusion-based backbone of the VTA model, we demonstrate the effectiveness of our VTA-LDM framework and conduct various ablations in the model design and training process. We also dive into several key aspects during the VTA generation, aiming at giving the community more insights about the model design and training process. Specifically, we concentrate on the vision encoder, auxiliary embeddings, and data augmentation techniques employed in the training process, and propose several significant suggestions that have been quantitatively validated through our experiments. A combination of these approaches can lead to a more powerful VTA backbone. We foresee the evolution of more realistic and natural video-audio generation models based on the insights gained from this study.

**Limitations & future work** The models presented in this paper are trained on a limited dataset, VGGSound, which primarily contains audios of single audio event data. Although we employ data augmentations and utilize a portion of the filtered YouTube test set, there is room for improvement. As part of our future work, we plan to build a more extensive, more diverse, and real-world-like high-quality dataset.

**Social impact** The capability to generate realistic audio from silent visual input can significantly enhance the accessibility of high-quality video-audio content, benefiting fields such as AI-generated content (AIGC) and virtual reality simulations. However, potential ethical concerns arise, including the possibility of misuse in creating deep fake videos or misleading audio content.

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
