# OpenReview forum: "Video-to-Audio generation with Hidden Alignment"
_ICLR.cc/2025/Conference — ICLR 2025 Conference Withdrawn Submission_

### Official Review · Reviewer_TAfS · 2024-10-29

**Soundness:** 1
**Presentation:** 2
**Contribution:** 2
**Rating:** 3
**Confidence:** 4

**Summary:**

This study empirically investigates how the performance of video-to-audio models would be affected by three algorithmic designs: vision encoders, additional usage of auxiliary conditions, and data augmentation techniques. The experimental results show that state-of-the-art performance can be achieved by choosing the best option in each design. They also bring several important observations: for example, the additional usage of auxiliary conditions helps the model to focus on the actual object producing the audio, and filtering training data boosts the audio-visual alignment while dropping the overall generation quality.

**Strengths:**

- The motivation behind this study is significant for a deeper understanding on how we should construct video-to-audio models. Many recent video-to-audio models are constructed by extending audio generation models to make them accept video (and any other auxiliary) conditions. Thus, the choice of vision encoders and auxiliary embeddings should be a particular and important challenge, which has empirically been explored in this study. Rigorous investigation would be expected to significantly benefit future research on video-to-audio technology.

- The baseline framework seems quite simple and appears straightforward to implement. The vision encoders and audio VAEs examined in this paper are all publicly available ones, which would greatly help the reproduction of the reported empirical results.

**Weaknesses:**

- Several descriptions in the paper are unclear, making it difficult to follow the authors' arguments.
  - Each vision encoder should have a different resolution (and provide a different size of features), but its details are not given in the paper. I would suggest that the authors provide a table or detailed description of the temporal and spatial resolutions used for each vision encoder, as well as how these were standardized (if at all) for fair comparison.
    - Particulaly, it would be beneficial to include the information related to temporal resolution of the features, because it should heavily affect the model's capability on temporal alignment. For example, in the standard setting, CAVP features would be extracted from 4-fps videos, while ImageBind features are from 1-fps videos.
  - Tables 1 and 4 show the evaluation results by multiple metrics, but it is not clear which metric means what, which makes it hard to properly understand the findings. For example, around l.392, the authors state that audio quality (maybe PAM) has been improved while generation quality (maybe FAD/IS) drops, but it is confusing due to the lack of the explanation on how these two kinds of quality differ.
  - The details of "Pretrain" in section 5.5 are missing. Actually, it may not align with the typical definition of data augmentation, which does not alter the main training objective.
  - The saliency maps shown in Fig. 2 seem interesting, but how to compute this saliency is not described.
  - The term "hidden alignment" in the title does not appear in the main text and is not clearly defined.

- The insights from the empirical results are somewhat limited.
  - In section 5.3, various encoders are examined in the experiments, but it is not clear which aspects of the vision encoder contribute the most to boosting the performance of the video-to-audio model, because they differ from each other in many ways such as resolution, model size, training dataset, pretraining method, performance on downstream tasks, etc. Empirically verifying this should be one of the most important insights for the choice of the vision encoder to be used in future video-to-audio studies. Simply speaking, what property of CLIP4clip contributed most to the performance gain?
  - A similar issue as above also exists in section 5.5. The authors argue that the data filtering leads to the most significant performance improvement, but, as it comprises two filtering processes based on semantic and temporal alignment, it is not clear which filtering contributes more. In addition, it is also not clear why the authors choose CLAP and AV-Align for the filtering process, though there are various other choices examined in section 5.3 (namely, ImageBind, LanguageBind, and CAVP).

**Questions:**

- See the first point in Weaknesses.
- Is the concatenation-based augmentation in section 5.5 the same one with that used in DiffFoley? If so, it is quite counter-intuitive that that augmentation does not boost the temporal alignment (actually drops it a bit in CAVP score).

---

### Official Review · Reviewer_kknc · 2024-11-02

**Soundness:** 2
**Presentation:** 4
**Contribution:** 1
**Rating:** 3
**Confidence:** 5

**Summary:**

The paper proposed a Video-to-Audio generation framework. The model consists of a pretrained audio VAE and vocoder, a latent-diffusion-model, and CLIP-based vision encoder. The paper studied different design choices, such as using different vision encoders, input different conditioning features, and different data augmentation methods during training.

**Strengths:**

1. Lots of studies on the design choices.
While many components of the model inherits previous pretrained frozen models (which I don't think is a weakness), the author has experimented with many versions of vision encoders (6 in the paper Table 2.).

2. Good paper writing.
The paper is well written. Despite a few missing details, the overall paper is easy to follow.

3. Lot's of evaluation metrics.
The paper has used lots of evaluate metrics, which help in assessing the overall model in semantics alignment and synchronization.

**Weaknesses:**

1. Lack of evaluation dataset and synchronization metrics
VGGSound is in fact a noisy dataset to measure both: on one hand, it contains overwhelming audio noises, and many video audio pairs are not semantically aligned; on the other hand, the video-audio synchronization is even poor due to camera shaking, out of scene sound sources, etc. Previous papers on VTA has evaluated on some other datasets, which can be simpler but much cleaner, e.g., evaluate semantics alignment on Landscapes dataset (Seeing&Hearing), or evaluate both semantics alignment and synchronization on AVSync15 (FoleyCrafter). Without evaluation on these datasets, the single-dataset (and also noisy dataset) evaluation results is pretty hard to convince the community.

The author explained in Sec. 5.2 to "omit alignment metrics, as they are tailored to produce shorter audio segments, which does not align with competitive standards". This reason is weak. Suppose the other datasets/metrics do have shorter audio-video segments, the author can simply extend the input video with black/white/the last frame, then after getting the audio, only retrieve the first several seconds' content. In fact, there are lots of videos in VGGSound that have frame sudden changes (some frames in one video are even not related to each other), therefore, the method I mentioned to deal with shorter video-audio segments is absolutely feasible.

2. Inconsistent / strange evaluation.

On VGGSound, the author mentioned in Sec. 4.1 that, on VGGSound, they are using 200k video for training and 3k for testing. Even without considering broken youtube links, VGGSound's original annotation file only contains 183,971 training videos and 15496 testing videos, summing up to no more than 200k. How come the author can use 203k VGGSound videos? Do the training and testing splits overlapping with each other?

Also, the evaluation metrics are inconsistent. For the same model and CLIP4CLIP encoder, the AV-Align metrics is reported as 0.243 in Table 1 but 0.225 in Table 2, while other metrics being the same. If this was a typo, I would consider it as very serious mistake.

3. Weaker performance compared to SOTA.
When comparing with other methods in Table 1, the author used CLIP4CLIP as the encoder. However, the proposed model, once equipped with ImageBind encoder (Note that ImageBind vision encoder uses frozen CLIP image encoder, which makes it the same as CLIP encoder in Table 1), the performance becomes much worse than some other methods. E.g., AV-Align is 0.189 compared to FoleyCrafter's 0.239, CLAP is 0.186 compared to IM2WAV's 0.306, PAM is 0.166, the worst among all compared models. That means, under fair comparison, the paper cannot achieve SOTA performance, as opposed to what is claimed.

Besides, even the proposed model is not stronger than previous methods, there should always be a more detailed comparison between the proposed framework and previous works. E.g., DiffFoley and FoleyCrafter also used LDM for generation, what is the difference between the proposed LDM and theirs?

4. Some missing but important details.

(1) In Sec. 3.3, when UNet attending the vision features, does each audio feature only see vision features within its corresponding time window? Or, each audio features is attended to all vision features, regardless its time? If it is the former, how exactly did you split the vision tokens to achieve time-aware attention? If it is the latter (every audio feature in UNet sees exactly the same vision encoded feature, regardless of time) and the model can still achieve good synchronization results, then the only explanation should be that, the vision features have already be implicitly encoded with time information (e.g., CLIP4CLIP might already have positional encodings on temporal dimension), which might be the reason why using PE does not help in Table 3? Any more discussions on this?

(2) How did you sample the 3k validation data? Did you first apply data alignment filter to get a clean subset first, then choose from it? Or did you simply random sample from original noisy VGGSound?

(3) There should be formal explanation on each metric you are using, at least in supplementary material. The current rough explanation in Sec. 4.2 is vague. For example, the author mentioned to use CAVP as semantics and synchronization evaluation metric. However, CAVP is only a contrastively trained model in DiffFoley. The original DiffFoley paper used a trained classifier to tell synchronization and semantics alignment instead, and called the metric "Align Acc". Did you use CAVP features just as in CLIPSim for evaluation, or did you use Align Acc? The detail is missing though.

**Questions:**

Following my weakness mentioned above, I would like to see answers to the following questions:

1. The proposed method's performance, and comparison, on Landscapes (Compare to Seeing&Hearing) and AVSync15(Compare to FoleyCrafter).

2. The author's explanation on strange training/testing splits of VGGSound, and inconsistent data in Table 1 & Table 2.

3. A conclusion on, under fair comparison, whether the proposed model is truly SOTA or not, and provide more concrete  comparison between the proposed model and others, as well as some analysis on why (or why not) the proposed model is SOTA.

4. The 3 questions I've asked in Last point in weakness.

---

### Official Review · Reviewer_7Cqr · 2024-11-06

**Soundness:** 3
**Presentation:** 3
**Contribution:** 2
**Rating:** 5
**Confidence:** 4

**Summary:**

This paper proposes a video-to-audio model by investigating vision encoders, auxiliary embeddings, and data augmentation techniques. It shows an SOTA performance.
It uses the CLIP4CLIP as a video encoder, LDM, to generate the mel-spectrogram space.
It uses VGGSound 200k videos for training. It also filters out the video-audio pairs with low similarity.

**Strengths:**

1.	This paper proposes a baseline of the LDM model that surpasses the other comparison methods in some metrics.
2.	It performed various ablations on video encoder selection, data augmentation, and text embedding, adding up to this paper's empirical value.

**Weaknesses:**

1.	It shows that the text embedding would help the model generate the audio; I think it makes sense, but I’d like to know how details of the caption are needed. Does dense video caption help, or just a succinct video caption is enough?
2.	Minor: Why do Experiment Setup and Experiments in separate sections? I think it’s common to pub experimental setup as a subsection under the experiment section.
3.	In L383, the data augmentation is to randomly combine video and audio segments. But does this reduce the alignment between video and audio?
4.	It seems the author did many kinds of data preprocessing but only evaluated them independently and expected the mixture of them to work the best. However, I think it should be within the scope of this paper to give a final combined setting of those data augmentation, which makes me feel the unreadiness of this paper.
5.	I found this paper a baseline with empirical ablation, but it is not clear where the novelty lands.

**Questions:**

1.	Is the LDM learned from scratch?
2.	The CLIP4CLIP feature is not clear to me. Does it use the spatial-temporal feature, or is it just a mean pooled feature vector, as indicated by the original paper?
3.	What is the position embedding in L338? Is it the video position embedding or text position embedding?
4.	I’d like to see the explanation on why the attention on video is more focused by adding text, as shown in Fig.3. I think this salience map is the UNet activation as query and the video patch feature as key.
5.	How to remove those videos where the background music is not that correlated with the video content?

---

### Note · Authors · 2024-11-13

**Comment:**

We would like to express our sincere gratitude to the reviewers and the ACs for their valuable time and feedback on our work. We have decided to withdraw the manuscript at this time. We remain committed to ongoing improvements and refinements to our research based on these advices.

**Withdrawal Confirmation:**

I have read and agree with the venue's withdrawal policy on behalf of myself and my co-authors.